# OpenReview forum: "Safeguarding Text-to-Image Generation via Inference-Time Prompt-Noise Optimization"
_ICLR.cc/2026/Conference — ICLR 2026 Conference Withdrawn Submission_

### Official Review · Reviewer_WADp · 2025-10-15

**Soundness:** 2
**Presentation:** 3
**Contribution:** 2
**Rating:** 4
**Confidence:** 3

**Summary:**

This paper addresses the critical challenge of unsafe content generation in Text-to-Image (T2I) diffusion models. The authors propose Prompt-Noise Optimization (PNO), a training-free, inference-time framework that jointly optimizes two core components in the DDIM sampling process.

**Strengths:**

1.By co-optimizing prompt embeddings (controlling semantics) and noise trajectories (controlling low-level details), PNO avoids the limitations of single-component methods—prompt-only optimization causes severe semantic deviation, while noise-only optimization fails to suppress toxic semantics.

2.On the Ring-a-bell dataset, baselines like SalUn reach 100% inappropriate output, whereas PNO maintains 0% inappropriate generation.

**Weaknesses:**

1.While framed as an optimization-based approach, PNO lacks key theoretical analysis. There is no formal proof of how the regularization term preserves Gaussian noise properties, nor discussion of convergence guarantees for the iterative optimization (Algorithm 2). This limits understanding of PNO’s reliability across edge cases.

2.The paper states PNO requires ≤20 seconds for 60% of prompts (≤3 iterations) and sets a maximum of 25 iterations for complex cases. This makes it hard to assess PNO’s suitability for low-latency scenarios.

3.Prompt alignment is measured solely via CLIP score, leaving PNO’s real-world usability untested.

4.PNO’s performance is heavily tied to pre-trained safety classifiers. The paper does not address PNO’s performance in low-resource scenarios.

**Questions:**

1.Provide formal convergence analysis for PNO’s iterative optimization, including the conditions for the "toxicity loss + regularization term" to converge to a minimum and how the regularization coefficient \lambda impacts convergence speed and noise Gaussianity.

2.Supplement latency data for different iteration counts (e.g., 10, 15, 25 iterations), clearly define "complex toxic prompts" (e.g., specific I2P categories, prompt complexity metrics), and provide their corresponding iteration count distribution.

3.Quantify the degradation of PNO’s toxicity suppression rate in low-resource scenarios (e.g., using noisy crowd-sourced labels for safety evaluators) and clarify whether multi-objective optimization (e.g., balancing safety and aesthetic quality) has been considered for integration into PNO.

4.Provide qualitative or quantitative results for prompts linking benign concepts to toxic elements to verify if PNO can suppress toxic elements while preserving core benign semantics.

---

### Official Review · Reviewer_wwZL · 2025-10-27

**Soundness:** 2
**Presentation:** 3
**Contribution:** 2
**Rating:** 2
**Confidence:** 4

**Summary:**

This paper proposes Prompt-Noise Optimization (PNO), a training-free method to mitigate unsafe image generation in T2I diffusion models. The core idea involves an inference-time optimization framework that jointly adjusts the continuous prompt embedding and the noise trajectory during the sampling process to minimize a toxicity score derived from a safety classifier.

**Strengths:**

1. The writing  is easy to understand
2. The method is training free.

**Weaknesses:**

1. Regarding L_toxic, the loss function is related to the Q16 score, and the final model evaluation metric is also Q16. This may lead the method to optimize solely for Q16, even though the final Q16 evaluation results are good, it does not necessarily mean the actual performance is sufficiently good. This appears to be merely a task-specific optimization of the evaluation metric.
2. Regarding L_reg, its specific expression form does not seem to be clear.
3. Increasing the optimization step size k appears to significantly prolong inference time, which undermines the method's usability.
4. The article only includes a small number of quantitative experiments on the latest diffusion models such as SD3 and Flux, lacking detailed quantitative and qualitative experimental results.

**Questions:**

See weaknesses

**Details Of Ethics Concerns:**

No.

---

### Official Review · Reviewer_EDZ7 · 2025-10-27

**Soundness:** 1
**Presentation:** 2
**Contribution:** 1
**Rating:** 2
**Confidence:** 4

**Summary:**

This paper proposes Prompt-Noise Optimization (PNO), a training-free inference-time framework that safeguards text-to-image diffusion models from generating unsafe or inappropriate content. PNO jointly optimizes the continuous prompt embedding and the diffusion noise trajectory during sampling, minimizing a toxicity loss from an image safety classifier while maintaining text–image alignment. Extensive experiments on I2P and Ring-a-Bell datasets show that PNO achieves state-of-the-art safety performance, strong robustness to adversarial prompts, and an optimal trade-off between image safety and prompt fidelity, all with minimal inference overhead.

**Strengths:**

- This paper introduces an inference-time joint optimization over prompt embeddings and noise trajectories, which requires no retraining or fine-tuning while effectively reducing unsafe content.
- The proposed method exhibits a superior Pareto frontier between toxicity reduction and CLIP-based alignment, enabling flexible control over safety–fidelity trade-offs with minimal computational overhead.
- The insight derived from Lemma 1 for designing the regularization term is quite good.
- The approach consistently outperforms existing safety mechanisms across multiple benchmarks (I2P, Ring-a-Bell), achieving near-100% safe generations and strong robustness against adversarial attacks.

**Weaknesses:**

- The paper primarily presents an empirical approach with limited theoretical justification. While the proposed optimization framework is experimentally validated, it lacks a clear analytical understanding of why the joint optimization of prompt embeddings and noise trajectories is effective.
- The method lacks genuine novelty, as it essentially combines two existing ideas, initial noise optimization [1, 2] and prompt optimization [3], without introducing new theoretical or algorithmic insights.
- Section 4.3 notes that optimizing in the CLIP embedding space is more stable and preserves semantics, but this is trivial and well-known from prior work.
- The toxicity loss comes from the Q16 classifier, so directly optimizing it can artificially improve the Q16 metric. Therefore, I focus on metrics beyond Q16. In Table 2, while PNO achieves the best performance on VLM IP, its CLIP Score and PickScore are relatively poor compared to other methods.
- The Ring-a-Bell adversarial test uses only 50 prompts, which is insufficient to support claims of robustness. A larger-scale evaluation (at least 100 prompts) would be needed to draw statistically meaningful conclusions.
- The paper omits ablation studies on different values of $\lambda$, the regularization weight, which is critical for understanding the trade-off between safety and generative fidelity.

[1] Wallace, Bram, et al. End-to-end diffusion latent optimization improves classifier guidance. CVPR 2023.
[2] Novack, Zachary, et al. DITTO: diffusion inference-time T-optimization for music generation. ICML 2024.
[3] Chung, Hyungjin, et al. Prompt-tuning Latent Diffusion Models for Inverse Problems. ICML 2024.

**Questions:**

- Could the authors include the optimization curves of toxic loss and reg loss during the optimization process?
- Did gradient explosion or vanishing occur during the optimization process, and if so, how was it addressed?
- Other questions are discussed in the **Weaknesses** section.

---

### Official Review · Reviewer_rrh4 · 2025-10-29

**Soundness:** 2
**Presentation:** 3
**Contribution:** 2
**Rating:** 4
**Confidence:** 4

**Summary:**

This paper introduces PNO, a training-free, inference-time optimization framework for mitigating unsafe content in T2I diffusion models. Instead of fine-tuning model parameters, PNO jointly optimizes the continuous prompt embedding and the noise trajectory during sampling to reduce a learned “toxicity” score from an image safety classifier. The method requires no retraining, maintains comparable inference efficiency, and demonstrates superior robustness to adversarial prompts. Experiments on I2P and Ring-a-Bell datasets show that PNO outperforms strong baselines.

**Strengths:**

1. The experiments are comprehensive, which cover multiple baselines and ablations.
2. The paper is well-written and easy to follow.

**Weaknesses:**

1. Motivation for Joint Optimization: Although the method jointly optimizes both the prompt embedding and the noise trajectory to achieve safe text-to-image generation, the motivation for such design is unclear. If the unsafe content is merely from the unsafe input prompt, why not simply filter out or revise these malicious prompt, but use more resources to optimize both prompt and noise? Moreover, if the prompt is unsafe, is it essential to make it the output image still aligned with the semantics of the prompt?
2. Cost inefficiency compared with baselines: PNO introduces significant inference-time complexity by optimizing a very large noise trajectory in addition to the prompt embedding. This iterative procedure is far more complicated than common defenses like prompt filtering or negative prompts. Prompt-only optimization does improve safety, and negative prompting already achieves moderate safety.
3. Limited number of prompts: The evaluation relies on a narrow and highly curated set of prompts. Specifically, the experiments use only 50 adversarial prompts from the Ring-a-Bell dataset, but it is easy to generate a much larger variety of adversarial prompts using modern LLMs. Thus, the current evaluation setup may overstate the method’s effectiveness and robustness.
4. Vulnerability to Adaptive Attacks and Evaluation Bias: It is unclear whether the proposed method can defend against adaptive attacks. If an attacker specifically targets the Q16 classifier used in the optimization loop, the entire pipeline could fail, as the method directly relies on Q16 to guide safety optimization. Moreover, using the same classifier both for optimization and for evaluation introduces potential bias and risks overestimating performance.

**Questions:**

See above

---

### Note · Authors · 2025-11-12

I have read and agree with the venue's withdrawal policy on behalf of myself and my co-authors.